# Physiological Responses of *Paulownia fortunei* to Leaf Herbivory by *Epicauta ruficeps*: Nitrogen Assimilation, Porphyrin Metabolism, and ROS-Driven Antioxidant and Phenylpropanoid Responses

**DOI:** 10.3390/plants14233659

**Published:** 2025-11-30

**Authors:** Fan Wang, Zhongke Lv, Lizhi Xiao, Bo Chen, Wenhuan Liu, Jiaqing Huang, Gaoqiang Liu, Yuchen Yan, Jianhua Huang, Guoqun Yang

**Affiliations:** 1Key Laboratory of Forest Bio-Resources and Integrated Pest Management for Higher Education in Hunan Province, Central South University of Forestry and Technology, Changsha 410004, China; 2School of Life Science and Technology, Central South University of Forestry and Technology, Changsha 410004, China; 3Key Laboratory of Cultivation and Protection for Non-Wood Forest Trees, Ministry of Education, Central South University of Forestry and Technology, Changsha 410004, China; 4Key Laboratory of Research and Utilization of Ethnomedicinal Plant Resources of Hunan Province, College of Biological and Food Engineering, Huaihua University, Huaihua 418099, China

**Keywords:** insect feeding, porphyrin metabolism, nitrogen assimilation, photosynthetic

## Abstract

*Paulownia fortunei* is an important economic tree species that possesses numerous biological and economic traits, such as fast growth, strong stress resistance, and excellent wood properties. The cultivation of this species is pervasive across numerous regions of China. *Epicauta ruficeps*, a common pest species of *P. fortunei*, typically consumes the foliage of its host plant. However, there are currently no reported studies on the physiological and biochemical mechanisms underlying *P. fortunei* response to *E. ruficeps* feeding. In this study, we discovered that the enhancement of nitrogen assimilation and porphyrin metabolism directly contributes to the maintenance of the steady state of photosynthetic activity in *P. fortunei* leaves. Meanwhile, *E. ruficeps* feeding also leads to an increase in the level of reactive oxygen species (ROS) in *P. fortunei* leaves. As key signaling molecules, the elevated level of ROS activates the antioxidant system and phenylpropanoid metabolism, which in turn results in increased antioxidant enzyme activity, as well as increased contents of antioxidants and lignin. The aforementioned changes have the potential to reduce the degree of membrane lipid peroxidation and enhance the mechanical strength of leaf tissues. Consequently, this can assist in maintaining the steady state of photosynthesis indirectly. In summary, the present study elucidates the physiological and biochemical mechanisms underlying the maintenance of the steady state of photosynthetic activity in *P. fortunei* after being feeded by *E. ruficeps* from multiple dimensions. Simultaneously, it lays a theoretical foundation and provides data support for the subsequent comprehensive analysis of the molecular mechanisms involved in *P. fortunei* response to *E. ruficeps* feeding.

## 1. Introduction

*Paulownia* spp. is native to China and is a tall deciduous tree belonging to the family Paulowniaceae and the genus Paulownia. It has a wide planting area in China and a long history of cultivation [1]. As an important economic tree species, Paulownia possesses various excellent biological characteristics and commercial value. Its leaves can secrete a sticky substance that adsorbs harmful gases and dust from the air, making it an ideal urban greening tree species [2]. Paulownia wood is lightweight, corrosion-resistant, highly fire-resistant, and stable; has good thermal insulation properties; and features an attractive grain [3]. It also encompasses good acoustic properties and is easy to process, making it widely used in construction, papermaking, furniture manufacturing, and instrument production [3]. In addition, Paulownia flowers are rich in active compounds that may have antibacterial and anti-inflammatory effects and help alleviate respiratory conditions [4]. Its nectar is also thought to have cosmetic benefits. *Epicauta ruficeps* belongs to the order Coleoptera and the family Meloidae and is widely distributed throughout East Asia, including China, Japan, and Korea [5,6]. The life cycle of the *E. ruficeps* is complex, consisting of four stages: egg, larva, pupa, and adult. The larvae are predatory and primarily feed on the eggs of insects such as locusts, making them a potential candidate for biological pest control [7,8,9]. However, the adult *E. ruficeps* are herbivorous and often gather in large numbers to feed on the leaves and flowers of various plants, which can classify them as a pest to some extent [10]. It is well known that adult *E. ruficeps* have a particular preference for feeding on Paulownia leaves [7]. However, current research has not reported any studies on the mechanisms of Paulownia responding to the feeding of *E. ruficeps*.

Significant progress has been made in understanding plant immune defense mechanisms against insect herbivory. Plants can recognize both physical damage caused by insect feeding and specific molecular patterns in insect oral secretions, termed Herbivore-Associated Molecular Patterns (HAMPs), such as fatty acid–amino acid conjugates (FACs) [11,12]. These HAMPs are identified as “non-self” signals by the plant immune system, triggering early signaling events—including Ca^2+^ influx, reactive oxygen species (ROS) bursts, and MAPK kinase cascades—that ultimately activate systemic defense responses [13,14]. Phytohormones play a central role in defense regulation, with jasmonic acid (JA), salicylic acid (SA), and ethylene (ET) serving as the most critical signaling molecules [15,16]. The JA signal typically governs defenses against chewing insects [11], while the SA signal primarily mediates responses to piercing-sucking insects [17]. The ET signal often synergizes with the JA signal to enhance resistance against chewing herbivores [15]. These signal pathways exhibit complex cross-talk, acting antagonistically or cooperatively to fine-tune defense responses. This ability to dynamically adjust defenses based on threat type and intensity is termed induced resistance, enabling plants to allocate resources efficiently against herbivory [18,19,20].

Fast-growing plants have advantages in resource use and ecological restoration because of their rapid growth and high biomass. Regarding their resistance to herbivores, two main ecological hypotheses have been proposed. The Resource Availability Hypothesis holds that, in resource-rich habitats, plants tend to allocate resources to rapid growth rather than to defense. As a result, fast-growing plants may exhibit lower levels of structural or chemical defenses and be more susceptible to herbivores, relying on rapid growth to compensate for damage. The Compensatory Growth Hypothesis, by contrast, proposes that these plants may display strong tissue regrowth and compensatory growth after damage, thereby mitigating the negative effects of herbivory on overall growth [21,22,23]. The two hypotheses have one thing in common: plant growth capacity is linked to its mitigating effect on herbivores. Photosynthesis is the core process by which plants acquire and store energy. It provides the raw materials and energy necessary for growth and metabolism. It is crucial for plant growth to maintain photosynthetic efficiency under stress is crucial for plant growth [24].

In general, insect feeding directly or indirectly suppresses photosynthetic activity. However, in our previous study we observed that when fed on by *E. ruficeps*, the leaves of *P. fortunei* maintained stable photosynthetic activity. Based on this observation, we further investigated the underlying physiological and biochemical mechanisms. This study aims to elucidate the intrinsic mechanisms enabling *P. fortunei* to sustain photosynthetic stability under *E. ruficeps* herbivory.

## 2. Materials and Methods

### 2.1. Plant Sample Selection and Collections

This study used wild *P. fortunei* individuals growing within the same forest stand as the research subjects. The sample was collected in June 2025 from the Taohualing sub-scenic area of Yuelu Mountain, which is located in Changsha City, Hunan Province, in southern China. Control samples were collected from leaves of healthy plants that showed no signs of insect feeding, while experimental samples were collected from leaves of plants that were confirmed by repeated field surveys to have experienced heavy feeding by *E. ruficeps*. The experiment comprised three groups, each consisting of three P. fortunei which were used as biological replicates. The trees were between 10 and 15 years old, and the experimental trees were scattered throughout the stand. The experiment was divided into three groups, with three Paulownia trees in each group. Based on the degree of leaf damage, the experimental trees were divided into two groups. In the T1 group, most leaves on each tree had multiple feeding holes but were structurally relatively intact, with less than 40% of the leaf area damaged. In the T2 group, most leaves on each tree were severely damaged, showing obvious structural defects, and more than 40% of the leaf area was damaged (Figure 1A). All samples were immediately frozen in liquid nitrogen in the field and stored at −80 °C in the laboratory until further use. Before determining physiological and biochemical parameters, tissue samples were ground to a fine powder in liquid nitrogen for subsequent analyses. oils were placed in sealed dark glass bottles and stored at 4 °C for subsequent research.

### 2.2. Determination of Chlorophyll Fluorescence Parameters

The chlorophyll fluorescence parameters were measured with a Mini-PAM chlorophyll fluorometer (WALZ, Effeltrich, Germany) on the sampled plants. Measurements were made after 9:00 PM, after the leaves were fully dark-adapted, five leaves were measured per treatment. Instrument settings were configured in the Win-Control 3 software according to the values reported by Peng et al. [25]. Measured parameters included initial fluorescence (*F*o), maximum fluorescence (*F*m), maximum photochemical efficiency (*F*v/*F*m), electron transport rate (*ETR*), photochemical quenching coefficient (*q*P), non-photochemical quenching coefficient (*q*N), effective quantum yield of PSII (*Y*(*II*)), non-regulated energy dissipation quantum yield (*Y*(*NO*)), and regulated non-photochemical quenching (*NPQ*). These indicators were directly exported from Win-Control 3 and used for subsequent analysis.

### 2.3. Determination of Photosynthetic Pigment Content

We weighed 0.5 g of tissue, placed it in 25 mL of 95% ethanol, and then incubated it at 4 °C in the dark for 24 h until the tissue was bleached. The extract was then centrifuged and the supernatant collected. The absorbance of the supernatant was measured at 665, 649, and 470 nm, and the contents of chlorophyll a (Chl a), chlorophyll b (Chl b), total chlorophyll (Chl a+b), and carotenoids (Car) were calculated according to the methods used by Jiang et al. [24].

### 2.4. Determination of Porphyrin Metabolism Markers

5-Aminolevulinic acid (ALA) was quantified essentially as described by Wu et al. [26]. Five grams of finely ground tissue was vortex-extracted for 2 min with 6 mL acetate buffer (pH 4.6). After centrifugation, 5 mL of the supernatant was mixed with 200 µL acetylacetone, heated at 100 °C for 10 min to drive condensation, cooled to room temperature, and then combined with an equal volume of freshly prepared Ehrlich reagent. Following a 15 min room-temperature incubation, absorbance was measured at 554 nm, and ALA content was calculated from a standard curve. Proto IX, Mg-Proto IX, and Pchlide were determined by the method of Wu et al. [26]. 0.3 g of tissue was extracted for 24 h at 4 °C in darkness in 25 mL of 80% alkaline acetone until the tissue blanched. After centrifugation, absorbances of the supernatant were recorded at 575, 590, and 628 nm, and the contents of the respective porphyrins were calculated with the equations provided by Liu et al. [27].

### 2.5. Determination of Other Physiological and Biochemical Indicators

The following physiological indicators were measured using assay kits purchased from Comin Biotechnology (Suzhou, China) and Sinobestbio Biotechnology (Shanghai, China): H_2_O_2_ content, superoxide (O_2_·^−^) production rate, superoxide dismutase (SOD) activity, catalase (CAT) activity, peroxidase (POD) activity, ascorbate peroxidase (APX) activity, glutathione peroxidase (GPX) activity, polyamine oxidase (PAO) activity, glutamine synthetase (GS) activity, glutamate synthase (GOGAT) activity, glutamate dehydrogenase (GDH) activity, glutathione content (GSH), ascorbic acid (AsA) content, proline (Pro) content, glutamate (Glu) content, total flavonoid content, and total lignin content [28]. Fresh leaf tissue samples that had been ground into powder under liquid-nitrogen conditions were retrieved, except for the measurements of total flavonoids and total lignin content. For each biological replicate of each indicator, 0.1 g of powder was weighed and placed into a centrifuge tube. Depending on the indicator, 1 mL of the extraction solution supplied with the corresponding kit was added. The samples were vortexed for 2 min, centrifuged, and the supernatant was collected as the test solution. All procedures were carried out according to the kit instructions. For measurements of total flavonoids and total lignin, after completing other analyses, dry the remaining unused tissue samples at 80 °C to constant weight, then regrind and pass through a 60-mesh sieve, for subsequent measurement. For each sample, weigh out 20 mg and 5 mg of tissue into two separate centrifuge tubes. Add 1 mL of the appropriate kit’s extraction solution to each tube to determine total flavonoids and total lignin, respectively. Perform all extraction and assay procedures strictly according to the kit instructions. Kit specific details are provided in Appendix A.

### 2.6. Data Analysis and Statistical Figure Plotting

Data were organized using Microsoft Excel 2021. Statistical analyses were performed using IBM SPSS Statistics version 22. One-way ANOVA (*p* < 0.05) was used to assess differences among groups, and post hoc multiple comparisons were conducted using the Waller–Duncan test. All statistical figures were generated with Origin 2021.

## 3. Results

### 3.1. Effects of E. ruficeps Feeding on P. fortunei Chlorophyll Fluorescence Parameters

The objective of this study, which measured multiple chlorophyll fluorescence parameters, was to investigate the effects of *E. ruficeps* feeding on *P. fortunei* photosynthetic performance. In typical circumstances, characterized by normal, unstressed growth conditions, plants generally exhibit an *F*v/*F*m ratio that exceeds 0.8. However, a very interesting phenomenon was discovered. In the aftermath of the gnawing by *E. ruficeps*, the *F*v/*F*m levels in both categories of damaged *P. fortunei* leaves did not decrease and remained above 0.8 (Figure 1B). It is noteworthy that the *F*v/*F*m in the more severely damaged T2 leaves was significantly higher than in the control. Conversely, the levels of *F*o and *F*m in both T1 and T2 leaves were found to be significantly lower than in the control (Figure 1C,D). Furthermore, *ETR*, *q*P, *q*N, *Y*(*II*), *Y*(*NPQ*), and *Y*(*NO*) in *P. fortunei* leaves exhibited no substantial alterations in response to *E. ruficeps* feeding (Figure 1E–G).

### 3.2. Effects of E. ruficeps Feeding on P. fortunei Nitrogen Metabolism

Nitrogen metabolism is a pivotal factor in plant growth and defense, regulating the supply of amino acids, proteins and secondary metabolites. The present study found that feeding by *E. ruficeps* strongly stimulated nitrogen metabolism in *P. fortunei*, with this effect being particularly pronounced in T2-type leaves. The activities of enzymes associated with the GS–GOGAT cycle exhibited a significant increase in T2-type leaves. GS, GOGAT and GDH activities exhibited a 126.30%, 172.86% and 184.21% increase, respectively, relative to the control (Figure 2A–D). Concurrently, *E. ruficeps* feeding induced elevated metabolic rates in T1-type leaves, as evidenced by significantly elevated GS and GDH activities in comparison to the control group. Conversely, GOGAT activity exhibited no substantial variation. Furthermore, the glutamate content was found to be significantly higher in both T1- and T2-type leaves in comparison to the control, with T2-type leaves exhibiting significantly higher levels in comparison to T1-type leaves (Figure 2E).

### 3.3. Effects of E. ruficeps Feeding on P. fortunei Porphyrin Metabolism

Glutmate, a nitrogenous metabolite, serves as the initial substrate for plant porphyrin metabolism, and the intensity of this metabolism depends to some extent on glutamate content (Figure 3A). In this study, we found that feeding by *E. ruficeps* increased glutamate levels in *P. fortunei* leaves, providing a potential material basis for enhanced porphyrin metabolism. Specifically, under *E. ruficeps* ganwing, ALA content was significantly increased in T2-type leaves of *P. fortunei* (*p* < 0.01), rising by 133.48% compared with control leaves (Figure 3B). The change in Proto IX content followed a similar trend to ALA; in T1-type leaves Proto IX was elevated relative to control but the change was not significant (*p* > 0.05) (Figure 3C). Downstream products of this pathway, Mg-Proto IX and Pchlide, showed particularly marked changes. In both T1- and T2-type leaves, Mg-Proto IX and Pchlide levels were significantly higher than in control (Figure 3D,E). Mg-Proto IX and Pchlide increased by 36.93% and 51.48% in T1-type leaves and by 65.11% and 65.83% in T2-type leaves.

### 3.4. Effects of E. ruficeps Feeding on the Photosynthetic Pigment Content of P. fortunei

Pchlide, as the initial substrate for chlorophyll biosynthesis, typically has a direct effect on downstream chlorophyll formation and the overall balance of photosynthetic pigments. Therefore, this study first examined the effects of *E. ruficeps* feeding on the main photosynthetic pigments (Chl a, Chl b, and Chl a+b) in *P. fortunei* leaves. The results indicate that *E. ruficeps* feeding had virtually no effect on chlorophyll content in *P. fortunei* leaves. Chl a, Chl b, and Chl a+b in T1 and T2 leaves showed only minimal changes compared with the control group, and none were statistically significant (*p* > 0.05) (Figure 4A–C). Interestingly, Car is a photosynthetic pigment not regulated by porphyrin metabolism, but it did show significant changes. Under *E. ruficeps* feeding, Car content in *P. fortunei* T1 and T2 leaves increased relative to the control group by 48.80% and 92.27%, respectively (Figure 4D).

### 3.5. Effects of E. ruficeps Feeding on the Antioxidant System of P. fortunei

When plants are subjected to stress, they typically trigger oxidative stress responses that activate antioxidant systems to scavenge excess ROS (Figure 5A). We found that under *E. ruficeps* feeding, *P. fortunei* leaves showed increases in both H_2_O_2_ content and O_2_·^−^ production rate, with the change in O_2_·^−^ production being the most pronounced. O_2_·^−^ production rates in T1- and T2-type leaves were significantly higher than the control, increasing by approximately 97.35% and 178.88%, respectively (*p* < 0.05) (Figure 5B). H_2_O_2_ content increased by 27.35% in T1-type leaves but this change was not statistically significant (*p* > 0.05), while in T2-type leaves it increased by about 58.20% and was statistically significant (*p* < 0.05) (Figure 5C). Regarding antioxidant enzymes, SOD and CAT activities in both T1- and T2-type leaves were significantly elevated (*p* < 0.05). GPX activity did not change significantly in T2-type leaves but was significantly increased in T1-type leaves, by about 103.85% relative to the control. Meanwhile, the non-enzymatic antioxidant GSH showed a change pattern consistent with GPX activity and was significantly elevated in both T1- and T2-type leaves (*p* < 0.05), by approximately 271.54% and 91.38%, respectively. In addition, *E. ruficeps* feeding led to significant increases in AsA content in T1- and T2-type leaves, by about 113.33% and 93.70%, respectively. POD activity showed no significant change (Figure 5F).

### 3.6. Effects of E. ruficeps Feeding on Proline Content, Polyamine Oxidase Activity, and Phenylpropanoid Metabolites of P. fortunei

To evaluate the impact of polyamine metabolism and the phenylpropanoid pathway on the response of *P. fortunei* to *E. ruficeps* feeding, we measured proline levels, PAO activity, and the amounts of total flavonoids and total lignins. The results showed that the proline content in both T1- and T2-type leaves increased significantly, by 32.52% and 67.36%, respectively, compared to the control (Figure 6B). However, PAO activity did not change significantly in either type of leaf (Figure 6A). Measurement of the two phenylpropanoid metabolites revealed that, upon *E. ruficeps* feeding, the total contents of lignin and flavonoids in T2-type leaves increased significantly by 214.47% and 71.46%, respectively, compared to the control. In T1-type leaves, total lignin and flavonoid content increased by 9.03% and 32.06%, respectively, but did not change significantly (Figure 6C,D).

## 4. Discussion

The coevolution of plants and insects is a protracted biological process in which plants have evolved intricate defensive strategies to counter herbivorous insect attacks [29]. These defensive mechanisms are typically highly dynamic, involving morphological, biochemical, and molecular adaptations designed to mitigate or prevent insect feeding. From an ecological-evolutionary perspective, plants confronting insect feeding can either “Resist” to reduce insect consumption or “Tolerate” to minimize the physiological costs of damage [30]. In this study, *P. fortunei* exhibited a distinct dual-track “Compensation–Defense” strategy. On one hand, it maintained photosynthetic pigment levels and repaired photosystem function through enhanced nitrogen metabolism and porphyrin metabolism. Conversely, it has been observed to fortify defenses against *E. ruficeps* feeding, a phenomenon attributed to the augmentation of ROS scavenging capacity and phenylpropanoid metabolism (Figure 7). This strategy is indicative of a resource allocation trade-off mechanism that has evolved over an extended period of coevolution with *E. ruficeps*. This mechanism ensures the maintenance of photosynthetic activity while allocating resources to invest in necessary defenses, thereby enhancing survival efficiency.

### 4.1. The Compensatory Response of P. fortunei to Feeding Pressure from E. ruficeps

The processes of plant growth, development, and stress resistance are closely intertwined with the metabolic processes of nitrogen. As a central node in nitrogen metabolism, glutamate biosynthesis assimilates inorganic nitrogen into organic nitrogen through the GS-GOGAT cycle [31]. This process supplies precursors for amino acid pools and directly couples carbon and nitrogen metabolic balance via α-ketoglutarate recycling [32]. Furthermore, the role of nitrogen metabolism as a regulatory hub for stress responses has been well-documented [33]. By dynamically modulating proline, glutathione, and chlorophyll biosynthesis, it synergistically maintains cellular redox homeostasis, osmotic balance, and photosynthetic activity to counteract adverse effects from stress factirs [34]. Concurrently, glutamate functions not only as a key amino acid for protein synthesis but also as the material foundation for porphyrin/chlorophyll biosynthesis [35]. A multitude of studies has substantiated that chlorophyll content and photosynthetic activity characteristically diminish in plants subsequent to insect feeding [36,37,38]. This decline in photosynthetic activity is a component of the intricate physiological response to herbivory, encompassing physical damage, metabolic shifts, and the activation of defense mechanisms [39,40]. The direct feeding of insects on foliage has been demonstrated to induce tissue damage, thereby reducing the effective surface area for photosynthesis [41]. Furthermore, studies have found that *Apolygus lucorum* infestation of jujube trees significantly reduces leaf Chl a and Chl b content, severely impairing chlorophyll metabolism and photosynthetic performance while upregulating gene expression associated with chlorophyll degradation [42]. Concurrently, piercing-sucking insects such as aphids have the capacity to puncture the plant’s phloem, thereby accessing the sap and directly damaging the vascular tissues [43,44]. This results in a substantial decrease in chlorophyll content, consequently hindering photosynthesis. For instance, piercing-sucking pests on tea plants penetrate the vascular tissues to feed on the plant’s sap, resulting in the curling of tea leaves and the development of dark brown or silvery spots, which lead to significant yield losses in tea production. In the event of an infection by *Arbordia hussaini*, a decline in chlorophyll levels is observed, resulting in a reduction in photosynthetic efficiency and subsequent leaf desiccation [45]. Infestation experiments on tomato leaves indicate that *Phenacoccus solenopsis* infection can inhibit photosynthesis in a manner that is not consistent with the expected relationship between parasite load and host response [46]. The degree of suppression may be dependent on the intensity of the infestation [46]. In this study, *P. fortunei* exhibited stable photosynthetic activity in the presence of *E. ruficeps*, largely attributable to enhanced nitrogen and porphyrin metabolism. This enabled damaged leaves to maintain or increase their *F*v/*F*m ratio during dark adaptation, thereby helping to maintain photosynthetic homeostasis. It is noteworthy that chlorophyll levels in *P. fortunei* remained stable despite enhanced nitrogen and porphyrin metabolism, likely due to coordinated chlorophyll biosynthesis and degradation. In addition to their role in photosynthesis as accessory pigments, carotenoids have been shown to protect excited-state chlorophyll through non-radiative energy dissipation and antioxidant activities [47]. When plants encounter external stressors, ROS are generated. At this juncture, carotenoids have the capacity to absorb excess light energy and quench singlet oxygen, thereby safeguarding chlorophyll from photo-oxidative damage. This process indirectly maintains photosynthetic stability by counteracting the effects of external stressors on chlorophyll [48]. In this study, we observed a significant increase in carotenoid levels in *P. fortunei*, which is clearly associated with the protection of photosynthetic activity.

### 4.2. The Defensive Response of P. fortunei to Feeding Pressure from E. ruficeps

Damage to plants caused by mechanical means, typically resulting from insect feeding, has been shown to induce oxidative stress [14]. This condition has been observed to result in a substantial accumulation of ROS [49]. In order to counteract oxidative damage, plants typically activate antioxidant systems [50]. This system consists of antioxidant enzymes and antioxidants. The elimination of ROS is achieved through the synergistic action of antioxidant enzymes and antioxidants [51]. In this study, we observed a significant increase in the activity of multiple antioxidant enzymes and antioxidant levels within the antioxidant system of *P. fortunei*, which correlates with the elevated ROS levels induced by *E. ruficeps* feeding. Moreover, the elevated levels of ROS within the plant are predominantly influenced by two factors. First is an imbalance in the photosynthetic electron transport chain [52]. Second is enhanced polyamine metabolism [53]. In instances where the photosynthetic electron transport chain is in an overly reduced state, there is an increased likelihood of electron leakage, which, in turn, can induce the production of substantial amounts of ROS [52]. In this study, the activity of PAO, a pivotal enzyme in polyamine metabolism, exhibited no substantial alteration, while ROS levels underwent an increase. This finding suggests a potential association between the elevated ROS levels induced by *E. ruficeps* feeding and the photosynthetic electron transport chain. The enhancement of the antioxidant system has been demonstrated to accelerate the scavenging of ROS in *P. fortunei*. This adaptive strategy indirectly contributes to the maintenance of photosynthetic activity. In addition, ROS themselves act as critical signaling molecules in defense responses, inducing the expression of defense genes and the synthesis of secondary metabolites [14]. In the context of *E. ruficeps* feeding, *P. fortunei* demonstrated a marked increase in flavonoid and lignin levels, two phenylpropanoid metabolites. This observation suggests that phenylpropanoid metabolism plays a pivotal role in mediating defense responses against *E. ruficeps*. It is noteworthy that the mechanisms of action for these secondary metabolites differ significantly. Flavonoids are generally considered to be vital antioxidants, predominantly engaged in the scavenging of ROS and the prevention of oxidative damage [54]. In contrast, lignin is predominantly linked to the enhancement of the mechanical strength of plant cell walls [51,55]. Consequently, we infer that phenylpropanoid metabolism functions in an indirect manner to regulate photosynthetic homeostasis.

Overall, this study reveals a dual “Compensation–Defense” response strategy employed by *P. fortunei* when subjected to *E. ruficeps* feeding. *P. fortunei* maintains or rapidly restores photosynthetic pigment and photosystem function by enhancing nitrogen metabolism and porphyrin metabolism. Simultaneously, *P. fortunei* elevates carotenoid content, activates multiple antioxidant enzymes, and enhances phenylpropanoid metabolism to scavenge ROS and strengthen mechanical rigidity, thereby indirectly sustaining photosynthetic homeostasis. These responses demonstrate both the adaptive resource allocation of plants toward herbivory pressure through long-term coevolution and the complex coupling between photosynthetic electron transport, nitrogen metabolism, and secondary metabolism.

## 5. Conclusions

In this study, we used integrated physiological and biochemical techniques to understand the mechanisms that maintain photosynthetic homeostasis in *P. fortunei* after *E. ruficeps* feeding. The results indicate that *P. fortunei* enhances nitrogen and porphyrin metabolism to provide essential substrates for chlorophyll synthesis and chloroplast repair. This process sustains or rapidly restores overall photosynthetic homeostasis. Concurrently, ROS elevation induced by *E. ruficeps* feeding serves as a key signaling molecule that triggers the activation of antioxidant systems and phenylpropanoid metabolism. This leads to increased carotenoid, flavonoid, and lignin content. These changes can be regarded as being conducive to maintaining the redox balance, enhance leaf mechanical strength, and indirectly protect photosynthetic function. Thus, our findings reveal a coupled regulatory mechanism among photosynthetic electron transport, nitrogen metabolism, and secondary metabolism. Our results also provide physiological evidence to help understand the resource allocation and defense trade-offs formed during the long-term coevolution of *P. fortunei* and *E. ruficeps*.

## Figures and Tables

**Figure 1 plants-14-03659-f001:**
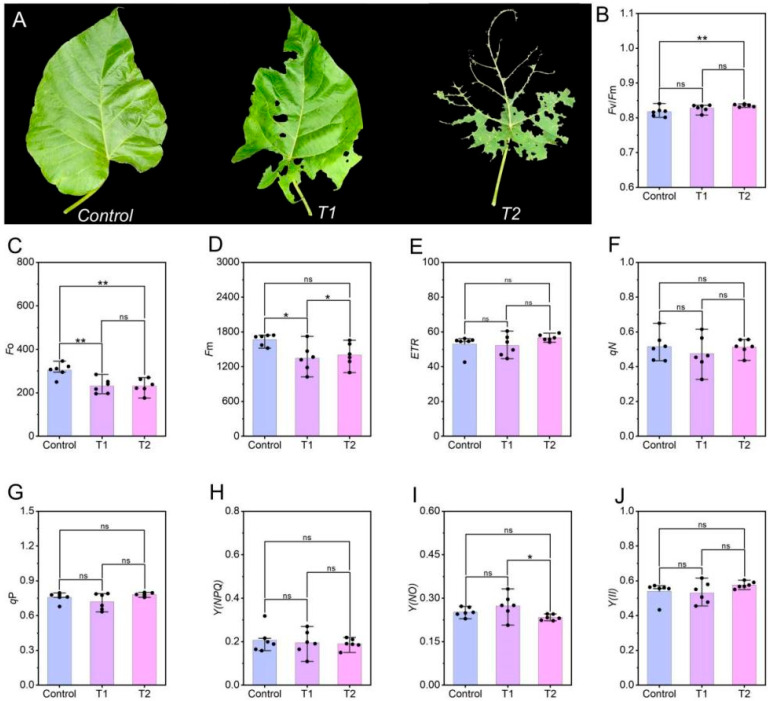
Leaf phenotypic changes and changes in chlorophyll fluorescence parameters resulting from feeding by *E. ruficeps*. (**A**) Leaf phenotypic characteristics. (**B**) Maximum photochemical efficiency, *F*v/*F*m. (**C**) Initial fluorescence, *F*o. (**D**) Maximum fluorescence, *F*m. (**E**) Electron transport rate, *ETR*. (**F**) Photochemical quenching coefficient, *q*P. (**G**) Non-photochemical quenching coefficient, *q*N. (**H**) Regulated non-photochemical quenching, *Y*(*NPQ*). (**I**) Non-regulated energy dissipation quantum yield, *Y*(*NO*). (**J**) Effective quantum yield of PSII, *Y*(*II*). Note: (ns), *p* > 0.05; (*), 0.01 < *p* < 0.05; (**), 0.001 < *p* < 0.01; (***), *p* < 0.001.

**Figure 2 plants-14-03659-f002:**
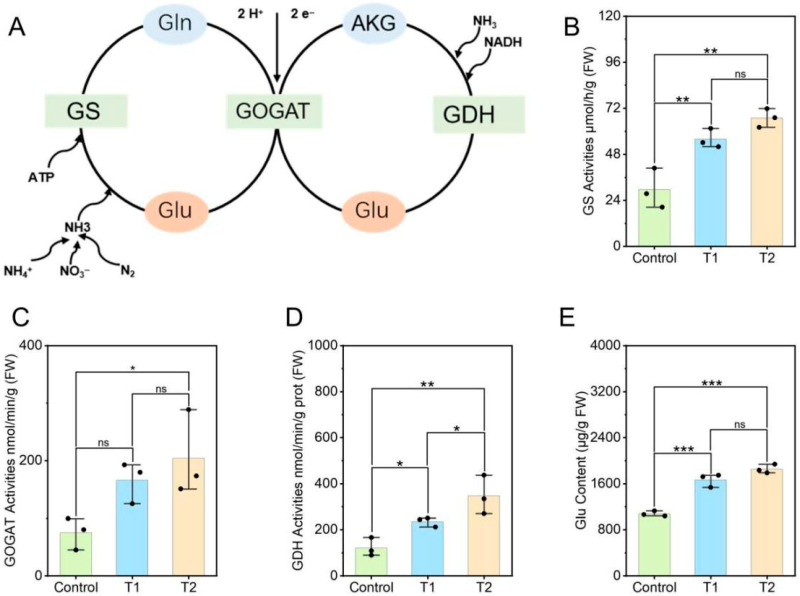
Effects of *E. ruficeps* feeding on nitrogen metabolism. (**A**) nitrogen metabolism pathway. (**B**) GS activities. (**C**) GOGAT activities. (**D**) GDH activities. (**E**) Glu content. Note: (ns), *p* > 0.05; (*), 0.01 < *p* < 0.05; (**), 0.001 < *p* < 0.01; (***), *p* < 0.001.

**Figure 3 plants-14-03659-f003:**
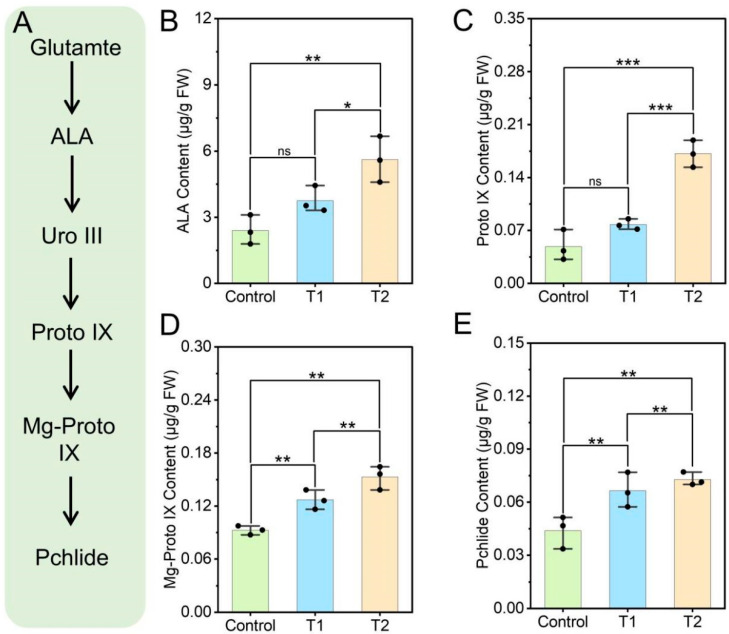
Effects of *E. ruficeps* feeding on porphyrin metabolism. (**A**) porphyrin metabolism pathway. (**B**) ALA content. (**C**) Proto IX content. (**D**) Mg-Proto IX content. (**E**) Pchlide content. Note: (ns), *p* > 0.05; (*), 0.01 < *p* < 0.05; (**), 0.001 < *p* < 0.01; (***), *p* < 0.001.

**Figure 4 plants-14-03659-f004:**
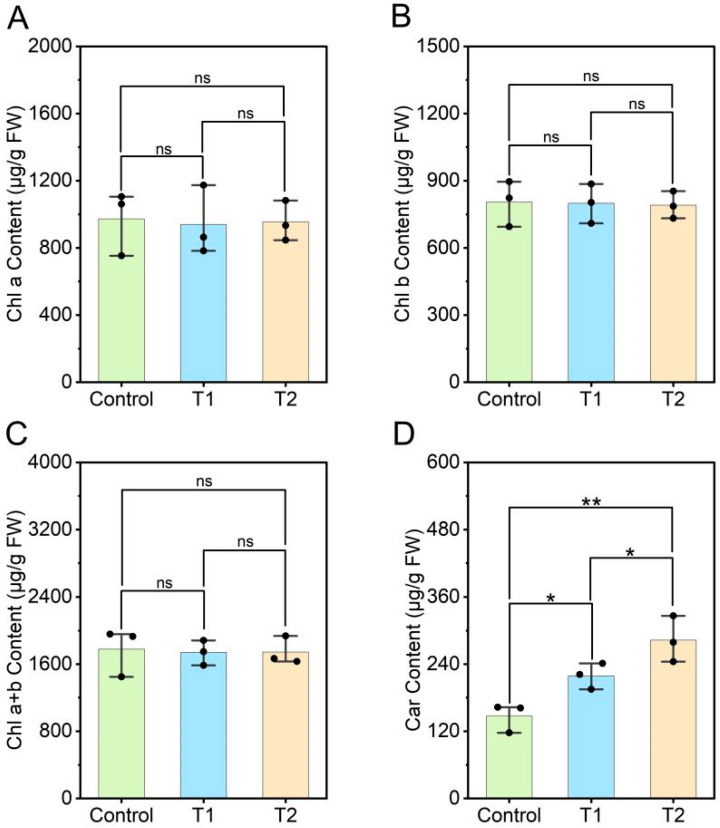
Effects of *E. ruficeps* feeding on photosynthetic pigment. (**A**) Chl a content. (**B**) Chl b content. (**C**) Chl a +b content. (**D**) Car content. Note: (ns), *p* > 0.05; (*), 0.01 < *p* < 0.05; (**), 0.001 < *p* < 0.01; (***), *p* < 0.001.

**Figure 5 plants-14-03659-f005:**
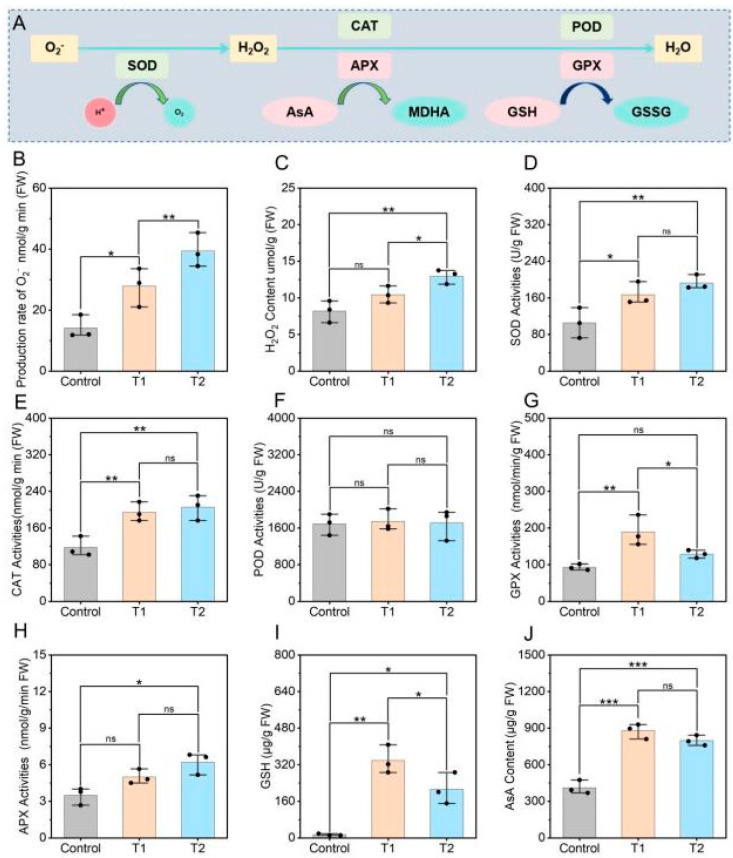
Effects of *E. ruficeps* feeding on antioxidant system. (**A**) Model diagram of the antioxidant system. (**B**) O_2_·^−^ production rate. (**C**) H_2_O_2_ content. (**D**) SOD activities. (**E**) CAT activities. (**F**) POD activities. (**G**) GPX activities. (**H**) APX activities. (**I**) GSH content. (**J**) AsA content. Note: (ns), *p* > 0.05; (*), 0.01 < *p* < 0.05; (**), 0.001 < *p* < 0.01; (***), *p* < 0.001.

**Figure 6 plants-14-03659-f006:**
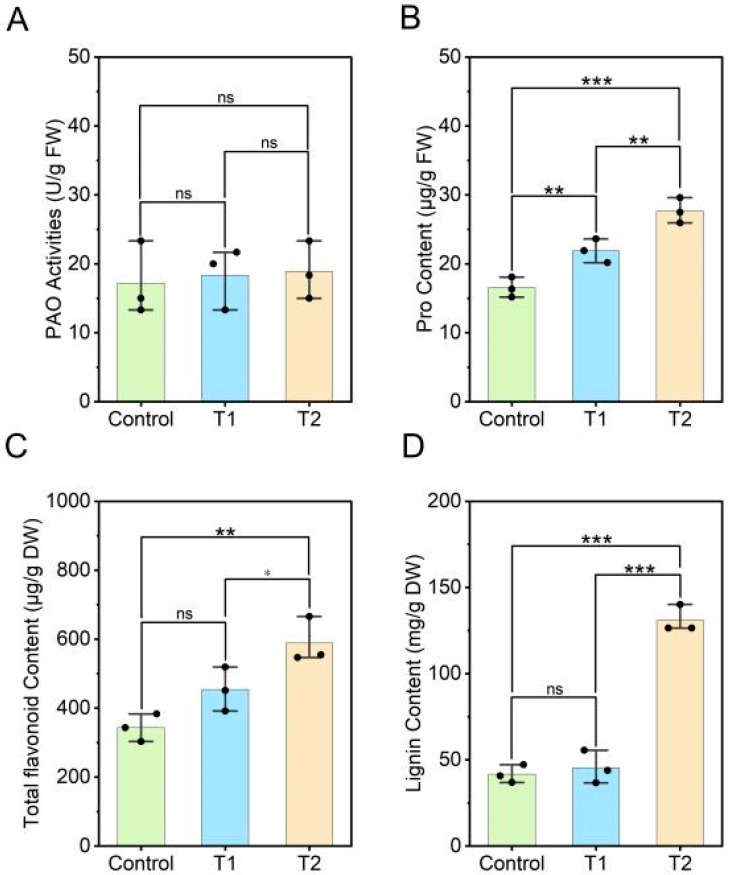
Effects of *E. ruficeps* Feeding on Proline Content, Polyamine Oxidase Activity, and Phenylpropanoid Metabolites. (**A**) PAO activities. (**B**) Pro content. (**C**) Total flavonoid content. (**D**) Lignin content. Note: (ns), *p* > 0.05; (*), 0.01 < *p* < 0.05; (**), 0.001 < *p* < 0.01; (***), *p* < 0.001.

**Figure 7 plants-14-03659-f007:**
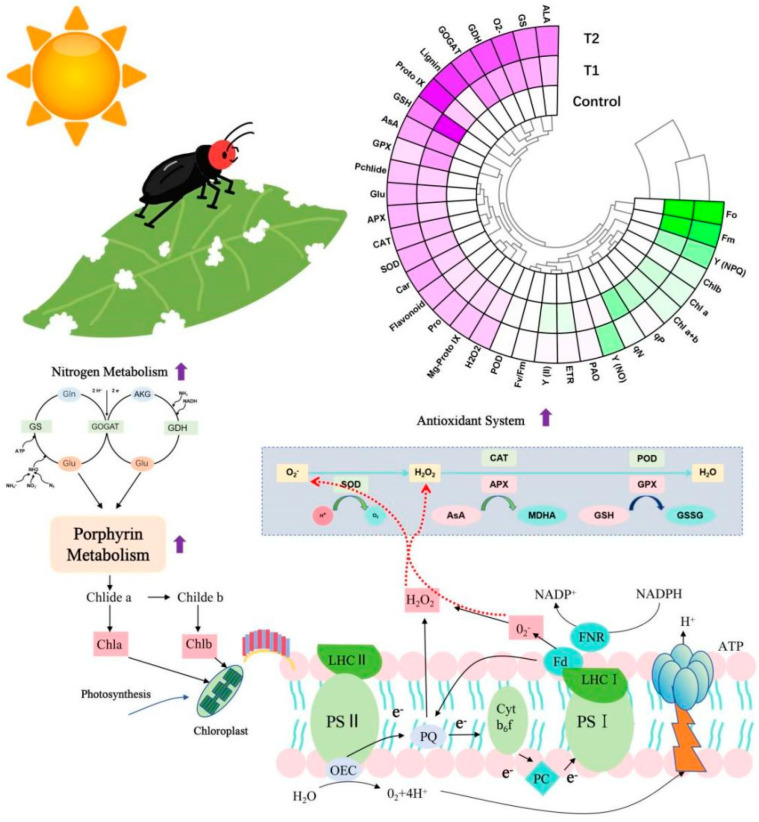
Schematic diagram of the primary physiological and biochemical mechanisms maintaining photosynthetic activity homeostasis in *P. fortunei*.

## Data Availability

Data are contained within the article.

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
