# Peer review of "Plants2025, 14(23), 3659;https://doi.org/10.3390/plants14233659"

_plants, 2025, doi:10.3390/plants14233659_

Round 1

Reviewer 1 Report

Comments and Suggestions for Authors

The manuscript “Physiological responses of Paulownia fortunei to leaf herbivory by Epicauta ruficeps: nitrogen assimilation, porphyrin metabolism, and ROS‑driven antioxidant and phenylpropanoid responses” have scientific worth and interesting results. This study provides insights into the physiological and biochemical responses of Paulownia fortunei to feeding by Epicauta ruficeps, highlighting the importance of nitrogen assimilation, ROS signaling, and antioxidant mechanisms in maintaining photosynthetic activity.

  •  Key Findings:
    • Paulownia fortunei is an economically important tree species with rapid growth and strong stress resistance.
    • Epicauta ruficeps is a common pest that feeds on the foliage of P. fortunei.
    • Enhanced nitrogen assimilation and porphyrin metabolism are crucial for maintaining photosynthetic activity in P. fortunei leaves.
    • E. ruficeps feeding increases reactive oxygen species (ROS) levels in P. fortunei leaves.
    • Elevated ROS levels activate the antioxidant system and phenylpropanoid metabolism, increasing antioxidant enzyme activity and lignin content.
    • These physiological changes help reduce membrane lipid peroxidation and enhance leaf tissue strength, indirectly supporting photosynthesis.

The findings lay the groundwork for further research into the molecular mechanisms underlying P. fortunei's response to pest infestations, contributing to improved management strategies for this economically valuable tree species.

Areas need to be improved (Reviewer comments)

  • Line 149-156: “2.5 Determination of other physiological and biochemical indicators: “H2O2 content, superoxide (O2· − ) production rate, superoxide dismutase (SOD) activity, catalase (CAT) activity, peroxidase (POD) ac-tivity, ascorbate peroxidase (APX) activity, glutathione peroxidase (GPX) activity, poly-amine oxidase (PAO) activity, glutamine synthetase (GS) activity, glutamate synthase (GOGAT) activity, glutamate dehydrogenase (GDH) activity, glutathione content (GSH), ascorbic acid (AsA) content, proline (Pro) content, glutamate (Glu) content, total flavonoid content, and total lignin content [27]” how much plant material was used? Fresh weight or dry weight? How were samples processed? How were the enzymatic activities calculated? Which standard was used and how, its concentration?
  • Line 159: “Total flavonoid and total lignin contents were determined last. Fresh sample powder was dried to a constant weight, and these measurements were carried out according to the kit instructions”.  Which standard was used to determine their concentration? Standard name and concentration. How much plant material was used to determine these parameters? What were the kit numbers? Lot number etc? manufacturers city and country?
  • Figure 1 to figure 6: which statistical analysis was used? Write this information in the legends, and what do the column bars mean and what do the dots stand for? Write this information in each legend.
  • Do italics all these scientific names used in the test and references for examples Line: 467.

Reviewer 2 Report

Comments and Suggestions for Authors

The manuscript has potential, but substantial improvements in writing, methodological clarity, analytical justification, and discussion structure are necessary before it can be considered for publication.

Round 2

Reviewer 2 Report

Comments and Suggestions for Authors

No further comments.